# Clinical signs of possible serious infection and associated mortality among young infants presenting at first-level health facilities

**Yasir Bin Nisar** [1]*, **Antoinette Tshefu**[2], **Adrien Lokangaka Longombe**[2], **Fabian Esamai**[3], **Irene Marete**[3], **Adejumoke Idowu Ayede**[4], **Ebunoluwa A. Adejuyigbe**[5], **Robinson D. Wammanda**[6], **Shamim Ahmad Qazi**[7], **Rajiv Bahl**[1]

1 Department of Maternal, Neonatal, Child and Adolescent Health and Ageing, World Health Organization, Geneva, Switzerland, 2 Department of Community Health, Kinshasa School of Public Health, Kinshasa, DR Congo, 3 Department of Child Health and Paediatrics, School of Medicine, Moi University, Eldoret, Kenya, 4 College of Medicine, University of Ibadan, University College Hospital, Ibadan, Nigeria, 5 Department of Paediatrics and Child Health, Obafemi Awolowo University, Ile-Ife, Nigeria, 6 Department of Community Medicine, Ahmadu Bello University Teaching Hospital, Ahmadu Bello University, Zaria, Nigeria, 7 Child and Newborn Health Consultant, Geneva, Switzerland

* nisary@who.int

## Abstract

### Background

The World Health Organization recommends inpatient hospital treatment of young infants up to two months old with any sign of possible serious infection. However, each sign may have a different risk of death. The current study aims to calculate the case fatality ratio for infants with individual or combined signs of possible serious infection, stratified by inpatient or outpatient treatment.

### Methods

We analysed data from the African Neonatal Sepsis Trial conducted in five sites in the Democratic Republic of the Congo, Kenya and Nigeria. Trained study nurses classified sick infants as pneumonia (fast breathing in 7–59 days old), severe pneumonia (fast breathing in 0–6 days old), clinical severe infection [severe chest indrawing, high (> = 38°C) or low body temperature (<35.5°C), stopped feeding well, or movement only when stimulated] or critical illness (convulsions, not able to feed at all, or no movement at all), and referred them to a hospital for inpatient treatment. Infants whose caregivers refused referral received outpatient treatment. The case fatality ratio by day 15 was calculated for individual and combined clinical signs and stratified by place of treatment. An infant with signs of clinical severe infection or severe pneumonia was recategorised as having low- (case fatality ratio ≤2%) or moderate- (case fatality ratio >2%) mortality risk.

### Results

Of 7129 young infants with a possible serious infection, fast breathing (in 7–59 days old) was the most prevalent sign (26%), followed by high body temperature (20%) and severe chest indrawing (19%). Infants with pneumonia had the lowest case fatality ratio (0.2%),

**Data Availability Statement:** All relevant data are within the paper and its Supporting Information files.

**Funding:** The study was funded by a grant from the Bill & Melinda Gates Foundation to WHO, which sponsored the study.

**Competing interests:** All other authors declare no competing interests. YBN and RB are WHO employees. The expressed views and opinions do not necessarily express the policies of the World Health Organization. This does not alter our adherence to PLOS ONE policies on sharing data and materials.

followed by severe pneumonia (2.0%), clinical severe infection (2.3%) and critical illness (16.9%). Infants with clinical severe infection had a wide range of case fatality ratios for individual signs (from 0.8% to 11.0%). Infants with pneumonia had similar case fatality ratio for outpatient and inpatient treatment (0.2% vs. 0.3%, p = 0.74). Infants with clinical severe infection or severe pneumonia had a lower case fatality ratio among those who received outpatient treatment compared to inpatient treatment (1.9% vs. 6.5%, p<0.0001). We recategorised infants into low-mortality risk signs (case fatality ratio ≤2%) of clinical severe infection (high body temperature, or severe chest indrawing) or severe pneumonia and moderate-mortality risk signs (case fatality ratio >2%) (stopped feeding well, movement only when stimulated, low body temperature or multiple signs of clinical severe infection). We found that both categories had four times lower case fatality ratio when treated as outpatient than inpatient treatment, i.e., 1.0% vs. 4.0% (p<0.0001) and 5.3% vs. 22.4% (p<0.0001), respectively. In contrast, infants with signs of critical illness had nearly two times higher case fatality ratio when treated as outpatient versus inpatient treatment (21.7% vs. 12.1%, p = 0.097).

## Conclusions

The mortality risk differs with clinical signs. Young infants with a possible serious infection can be grouped into those with low-mortality risk signs (high body temperature, or severe chest indrawing or severe pneumonia); moderate-mortality risk signs (stopped feeding well, movement only when stimulated, low body temperature or multiple signs of clinical severe infection), or high-mortality risk signs (signs of critical illness). New treatment strategies that consider differential mortality risks for the place of treatment and duration of inpatient treatment could be developed and evaluated based on these findings.

## Clinical trial registration

This trial was registered with the Australian New Zealand Clinical Trials Registry under ID ACTRN 12610000286044.

## Introduction

An estimated 2.4 million neonatal deaths occurred globally in 2019, representing 47% of under-five deaths [1]. The neonatal period in the first 28 days of life after birth is the most vulnerable time for survival. Each day 6700 neonatal deaths occur, one-third on the first day of birth, and three-quarters in the first week of life [1]. The global neonatal mortality rate decreased by 52% from 37 deaths per 1000 live births in 1990 to 17 deaths per 1000 live births in 1990 [1]. Bacterial infections, including sepsis, meningitis and pneumonia, account for 37% of neonatal deaths in the low-resource settings of South Asia and sub-Saharan Africa [2]. An estimated 9.8% died of the 6.9 million cases of possible serious bacterial infection (PSBI) in neonates and young infants up to two months of age in 2012 [3]. A recent systematic review of 26 studies from 14 countries estimated the neonatal sepsis incidence as 2824 per 100,000 live births with a mortality of 17.6%. However, the review also highlighted the variations in the definition of sepsis [4].

Early diagnosis requires recognition of clinical signs of bacterial sepsis [5–7]. Serious bacterial and viral infections are difficult to differentiate clinically in neonates and young infants up

to two months of age. World Health Organization (WHO) recommends that young infants with any sign of possible serious infection should be managed at a hospital with injectable antibiotics and supportive care [8]. Unfortunately, in many low-resource settings, families may not accept referral advice, and sick young infants are not taken to hospital [9–12]. Common barriers for not accepting referral advice are distance to the hospital, availability of transport and other logistical reasons such as lack of child care, cost of travel and treatment, lack of permission from the husband or family elders, concerns around quality of care, a previous negative experience or poor attitudes of health workers at the hospital, cultural or religious beliefs and the absence of referral protocols/algorithms [13–17]. WHO launched a guideline for managing PSBI in young infants when referral is not feasible in 2015 [18], based on evidence from high-quality randomised controlled trials [19–22]. The WHO guideline recommends that if a referral is not feasible, young infants with signs of clinical severe infection can be managed with a simplified antibiotic regimen of injectable gentamicin plus oral amoxicillin on an outpatient basis [18]. The critically ill young infants should be treated at a hospital because they require other supportive care apart from antibiotics, and most will be unable to take an oral antibiotic. In contrast, those infants aged 7–59 days with only fast breathing should be treated with oral amoxicillin on an outpatient basis without referral [18].

The updated WHO Integrated Management of Childhood Illness (IMCI) chart booklet for young infants [23] based on WHO guideline [18], recommends seven signs for urgent referral to hospital: not feeding well/not able to feed at all; convulsions; fast breathing; severe chest indrawing; high body temperature ($\geq$ 38˚C); low body temperature ($<$ 35.5˚C); movement only when stimulated or no movement at all [23]. WHO recommendations [18, 23] assumed a similar mortality risk for all signs of possible serious infection. However, individual signs may have variable mortality risks, which may have implications for treatment strategies [24]. Therefore, we conducted an analysis using the African Neonatal Sepsis Trial (AFRINEST) [19, 20] data to calculate the case fatality ratio (CFR) for young infants with individual or combined signs of possible serious infection, stratified by inpatient or outpatient treatment.

## Methods

### Study data

We used AFRINEST [19, 20] data for this analysis. Young infants less than two months of age were classified based on the updated IMCI chart booklet [23] as pneumonia [fast breathing (respiratory rate $\geq$ 60 breaths per minute) in 7–59 days old], severe pneumonia (fast breathing in 0–6 days old), clinical severe infection [stopped feeding well, severe chest indrawing, high body temperature ($\geq$ 38˚C), low body temperature ($<$ 35.5˚C), or movement only when stimulated] or critical illness (convulsions, not able to feed at all, or no movement at all). The study was conducted at one site, each in the Democratic Republic of Congo (DRC) and Kenya, and three sites in Nigeria (Ibadan, Ile-Ife and Zaria) and followed the same protocol, quality control, and coordination. Trial design and methods of AFRINEST have been described in detail elsewhere [25–27]. Briefly, young infants with any sign of infection were identified by trained community-level health workers (CLHW) in the community and referred to first-level health facilities in the study catchment area for further management. At the first-level health facilities, nurses trained in IMCI assessed infants for signs of possible serious infection. All young infants with any sign of possible serious infection were referred to a hospital. Those infants whose caregivers accepted referral to a hospital were followed up on day 15 of the initial assessment to collect information on their survival status. Those whose caregivers refused referral were offered simplified antibiotic treatment on an outpatient basis at the first-level health facility after consent was obtained and followed until day 15 to collect their survival status [19, 20].

In the DRC, the distance between the health centre (primary health care facility) and the Health Zone General Referral Hospital is approximately 50–55 kilometres (km) in the study area. Since there is no public transportation system available in the study area, people typically walk or ride a bicycle to reach the hospital. The time to reach the hospital can take up to 12 hours by foot or seven hours by bicycle, depending upon the weather, as in the rainy season, it takes a much longer time. In Kenya, the distance between the health centres and the district hospital is around 30 km in the study area. It usually takes several hours on foot and about 50 minutes by a motorized vehicle. In Ibadan, Nigeria, the secondary level referral hospital is approximately 40 km from the Primary Health Centres (PHC) in the study area. It usually takes 90 minutes to reach the hospital through public transport. In Ile-Ife, Nigeria, the referral hospital is situated 50 km away from the PHCs in the study area. The duration of travel time varies from 30 minutes to an hour, depending on the mode of transportation. In Zaria, Nigeria, the Gambo Sawaba General Hospital is at a distance of 15 km from the PHCs in the study area and the Ahmadu Bello University teaching hospital is over 30 km away. Most families walk, while only a few have the privilege of using motorized vehicles as most feeder roads are not motorable, especially during the rainy season. For this analysis, we selected 7129 young infants who presented at the first-level health facilities in the study catchment area, and IMCI trained nurses classified these infants as pneumonia, severe pneumonia, clinical severe infection, or critical illness.

## Outcomes and exposure

The study outcome was the survival status of a young infant with any sign of possible serious infection by day 15 after an initial assessment. CFR was defined as the number of deaths divided by the number of young infants with a specific clinical sign(s) by day 15. The main exposure variable was the clinical sign(s) at the time of presentation. Clinical signs considered in this analysis were fast breathing only, signs of clinical severe infection alone or in combination, or signs of critical illness alone or a combination [18, 23]. The place of treatment was categorised into either inpatient or outpatient.

## Statistical analysis

Stata 14.2 (Stata-Corp, College Station, TX, USA) was used for analysis. We calculated the prevalence of each clinical sign–alone or in combination—defined as the number of young infants with a specific clinical sign(s) divided by the total number of young infants with any sign of possible serious infection, expressed as a percent. CFR is reported with a 95% confidence interval (CI), calculated for IMCI classification of infants with possible serious infection as well as for each clinical sign–alone or in combination. Additionally, the CFR for IMCI classification and each clinical sign–alone or in combination, was compared between those who received outpatient and those who received inpatient treatment. Based on CFRs, infants with signs of clinical severe infection or severe pneumonia were recategorised for mortality risk into two groups, those who had low-mortality risk signs (defined as CFR of ≤ 2% for any sign), or those with moderate-mortality risk signs (defined as CFR of > 2% for any sign). The CFR for infants with signs of clinical severe infection or severe pneumonia with a low- or moderate-mortality risk sign was calculated and compared between outpatient and inpatient treatment and between the two age groups (0–6 days and 7–59 days) also. A Chi-square test was used to compare the CFR between outpatient and inpatient treatment groups, and p-values were reported. Cases with missing information about the survival status were excluded from the analysis. Infants were classified as outpatient or inpatient treatment group based on the initial treatment they received.

## Consent and ethical approval

The protocol was approved by the local institutional review boards at each site and by the WHO Ethical Review Committee. We obtained the site-specific ethics approvals from the University of Kinshasa School of Public Health Ethics Committee, DRC; the Moi University and Moi University Teaching Hospital Institutional Research and Ethics Committee, Eldoret, Kenya; the University of Ibadan/University College Ibadan Hospital Ethics Committee, Ibadan, Nigeria; the Obafemi Awolowo University Teaching Hospitals Complex Ethics Committee, Ile-Ife, Nigeria and the Ahmadu Bello University Teaching Hospital Ethics Committee, Zaria, Nigeria before the enrolment took place. Written and witnessed informed consent was obtained from parents/caregivers of infants before recruitment.

## Results

IMCI-trained nurses at the first-level health facilities assessed 18420 young infants 0–59 days old who were identified and referred by the CLHWs from the community at all five study sites. Of these, 7129 (38.7%) infants had any sign of possible serious infection, and they were followed up to day 15 after the initial assessment to document their survival status. The mean age of young infants with possible serious infection was 14.5 days (standard deviation 13.6). Out of 7129 young infants who had any sign of possible serious infection, 6040 (84.7%) were less than 28 days, and 3893 (54.6%) were males. We excluded 83 young infants from the analysis for whom information on survival status was missing. Thus data for 7046 young infants were analysed for CFR. About 89% (6269/7046) of young infants received outpatient treatment.

Amongst 7129 sick young infants, fast breathing (in 7–59 days old) was the most prevalent sign (26%), followed by high body temperature (20%) and severe chest indrawing (19%). More than one sign of clinical severe infection was reported in 8%, while signs of critical illness showed a relatively low prevalence (≤ 1%) in young infants (Fig 1).

The prevalence of low body temperature sign was 10 times higher in young infants 0–6 days of age (6.8%) than 7–59 days olds (0.7%). The prevalence of severe chest indrawing was 2.4 times higher in young infants 7–59 days old (23.8%) than those 0–6 days olds (10.5%) (Table 1).

Infants with pneumonia had the lowest CFR (0.2%), while those with any sign of critical illness had the highest CFR (16.9%) (Table 2). In terms of individual signs of clinical severe infection, infants with high body temperature had the lowest CFR (0.8%), whereas low body temperature had the highest CFR (11.0%). The CFRs of infants with any sign of critical illness ranged from 11.3% to 27.3% (Table 2).

Compared to inpatient hospital treatment, young infants who received outpatient treatment had a CFR 1.5 times lower for pneumonia (p = 0.74), 3.4 times lower for clinical severe infection or severe pneumonia (p < 0.0001) but 1.8 times higher for critical illness (p = 0.097) (Table 3). Infants with low-mortality risk signs of clinical severe infection or severe pneumonia had a CFR of 1% with outpatient treatment versus 4% with inpatient treatment (p<0.0001). Infants with moderate-mortality risk signs of clinical severe infection had a CFR of 5.3% with outpatient treatment versus 22.4% with inpatient treatment (p<0.0001) (Table 3).

The CFR was higher for both 0–6 days and 7–59 days age groups among severe pneumonia and clinical severe infection categories in inpatient versus outpatient treatment groups (Table 4). In contrast, among 0–6 days old infants with a critical illness, the CFR was more than twice in the outpatient treatment group compared to the inpatient treatment group. In the 0–6 days old age group, both low- and moderate-mortality risk signs had over twice the CFR for those who received inpatient treatment than those who received outpatient treatment. In contrast, CFR was nine times higher in 7–59 days old with moderate-mortality risk signs

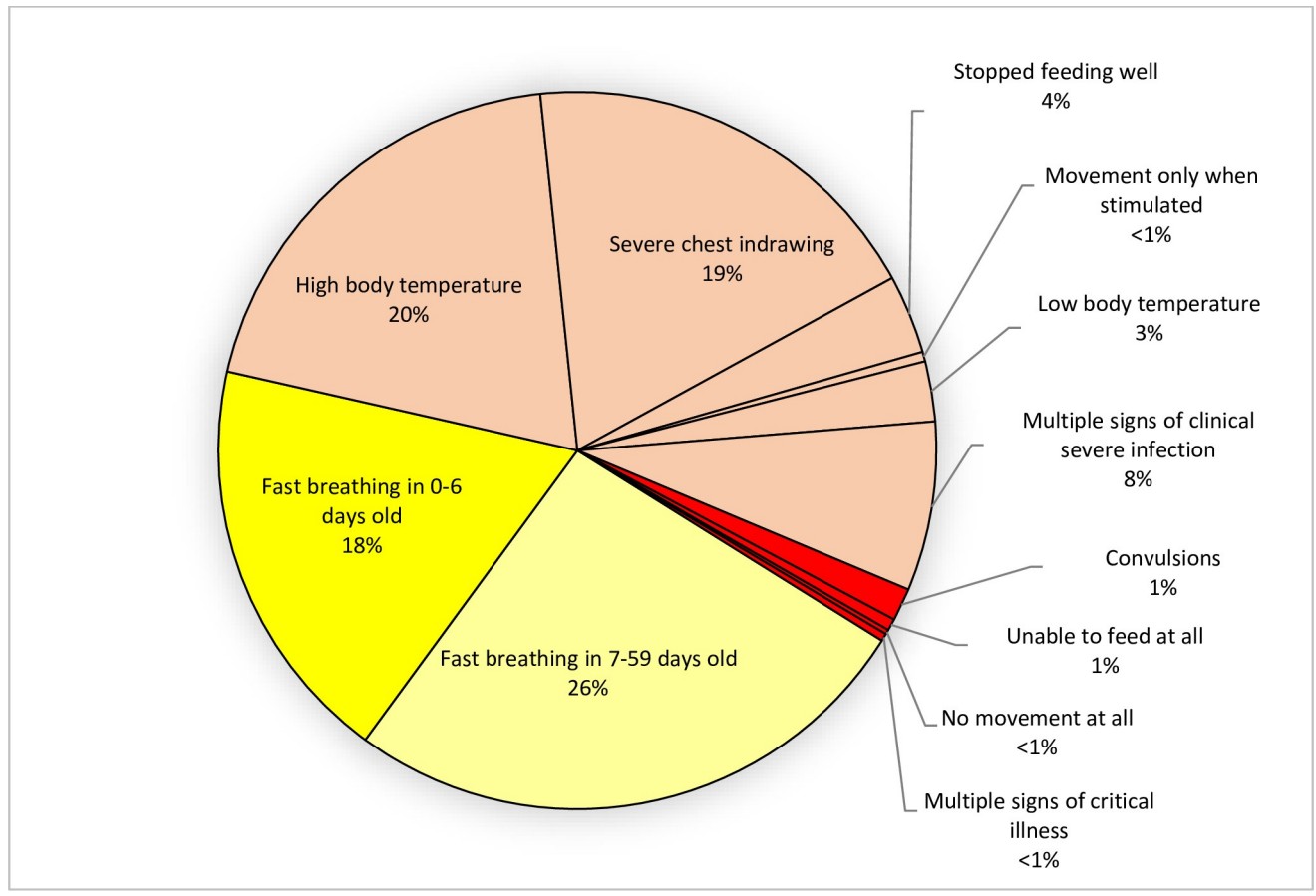

**Fig 1. Prevalence of clinical signs of possible serious infection in young infants 0–59 days old (n = 7129).**

amongst those who received inpatient treatment than those who received outpatient treatment, although the numbers are relatively small (Table 4).

We did not see any significant differences in the study sites separately in the outpatient and inpatient treatment groups by IMCI categories or by mortality risks (S1 Table).

## Discussion

### Key findings

Four key findings emerged from our analysis. First, infants with pneumonia (fast breathing in 7–59 days old) had the lowest CFR, and it was almost the same whether they received either outpatient or inpatient treatment. Second, infants with severe pneumonia had a relatively higher CFR than pneumonia but lower than those with a clinical severe infection or critical illness. Third, infants with any sign of critical illness had the highest CFR, and those treated as outpatients had it nearly twice as high compared to the ones treated as inpatients. Finally, in infants with signs of clinical severe infection, the CFR for individual signs showed substantial variation. This variation provided us with an opportunity to recategorise young infants with clinical severe infection or severe pneumonia into those with low- or moderate-mortality risk signs. Both categories had four times lower mortality among those who received outpatient treatment compared to inpatient treatment. This finding could be due to several reasons, such as delay in reaching the hospital after accepting referral advice, relatively sicker patients accepting referral

**Table 1. Distribution of signs of IMCI\* classification stratified by age categories in young infants presenting with any sign of possible serious infection (n = 7129).**

| IMCI* classification | Age 0–6 days (N = 2752) n (%) | Age 7–59 days (N = 4377) n (%) |
|---|---|---|
| **Pneumonia** | | |
| Fast breathing† in 7–59 days old‡ | Not applicable | 1866 (42.6%) |
| **Severe pneumonia** | | |
| Fast breathing† in 0–6 days old‡ | 1317 (47.9%) | Not applicable |
| **Clinical severe infection§** | | |
| High body temperature (≥ 38˚C) ‡ | 601 (21.8%) | 811 (18.5%) |
| Severe chest indrawing‡ | 290 (10.5%) | 1041 (23.8%) |
| Movement only when stimulated‡ | 12 (0.4%) | 19 (0.4%) |
| Stopped feeding well‡ | 83 (3.0%) | 170 (3.9%) |
| Low body temperature (< 35.5˚C) ‡ | 162 (6.9%) | 31 (0.7%) |
| Multiple signs of clinical severe infection§ | 187 (6.8%) | 359 (8.2%) |
| **Critical illness¶** | | |
| Convulsions‡ | 62 (2.2%) | 43 (1.0%) |
| Unable to feed at all‡ | 14 (0.5%) | 24 (0.5%) |
| No movement at all‡ | 8 (0.3%) | 7 (0.2%) |
| Multiple signs of critical illness¶ | 16 (0.6%) | 6 (0.4%) |

*IMCI: Integrated management of childhood illnesses.

† Fast breathing is defined as a respiratory rate of ≥ 60 breaths per minute.

‡ Young infant presenting with a single sign.

§ Clinical severe infection is defined as the presence of any one of the following signs: severe chest indrawing, high body temperature (≥ 38˚C), low body temperature (< 35.5˚C), stopped feeding well, or movement only when stimulated.

¶ Critical illness is defined as the presence of any one of the following signs: convulsions, unable to feed at all, or no movement at all.

advice, delay in getting appropriate treatment after reaching the hospital or inadequate quality of care, and development of hospital infections [16, 28–30]. These findings add support to the body of evidence suggesting a need to further improve the management of sick young infants in low resource settings where the quality of care and referral feasibility are a problem [15, 31–35].

Like the AFRINEST, two simplified antibiotic therapy trials (SATT) were conducted in Bangladesh and Pakistan. These trials showed that the simplified antibiotic regimen for young infants with clinical severe infection when referral to a hospital is not feasible was as efficacious as the standard regimen [21, 22]. AFRINEST and SATT studies [19–22] contributed evidence to the WHO guideline for managing PSBI in young infants when a referral is not feasible [18], but they did not report mortality for individual or combination of clinical signs. Hibberd et al. [24] reported CFR for some individual signs of PSBI in young infants in a multi-centre study from low resource settings; 0.8% for high fever, 5.7% for a breathing problem, 15.3% for feeding problem/stopped suckling/feeding, 18% for convulsions and 18.4% for hypothermia. However, they did not calculate it for all signs of infection and did not report it for the place of treatment [24]. In contrast, we reported mortality for individual clinical signs and compared it both for outpatient and inpatient treatment.

## Research and policy implications

These findings have important research and policy implications for developing and evaluating new management strategies for young infants aged 0–59 days with any sign of possible serious infection. First, infants with any sign of critical illness cannot be safely managed on an

**Table 2. Number of deaths, cases and case fatality ratio (CFR) of sick young infants by IMCI* classification [23] (n = 7046).**

| IMCI* classification of sick infants | Deaths/young infants | CFR[†] % (95% CI) |
|---|:---:|:---:|
| **Pneumonia** | | |
| Fast breathing[‡] in 7–59 days old[§] | 4/1843 | 0.2 (0.1, 0.6) |
| **Severe pneumonia** | | |
| Fast breathing[‡] in 0–6 days old[§] | 26/1291 | 2.0 (1.3, 2.9) |
| **Clinical severe infection[¶]** | **85/3746** | **2.3 (1.8, 2.8)** |
| High body temperature ($\geq$ 38˚C) [§] | 11/1409 | 0.8 (0.4, 1.4) |
| Severe chest indrawing[§] | 12/1329 | 0.9 (0.5, 1.6) |
| Movement only when stimulated[§] | 1/31 | 3.2 (0.1, 16.7) |
| Stopped feeding well[§] | 10/251 | 4.0 (1.9, 7.2) |
| Low body temperature ($<$ 35.5˚C)[§] | 20/182 | 11.0 (6.8, 16.5) |
| Multiple signs of clinical severe infection[¶] | 31/544 | 5.7 (3.9, 8.0) |
| **Critical illness[#]** | **28/166** | **16.9 (11.5, 23.4)** |
| Convulsions[§] | 11/97 | 11.3 (5.8, 19.4) |
| Unable to feed at all[§] | 8/35 | 22.9 (10.4, 40.1) |
| No movement at all[§] | 3/12 | 25.0 (5.5, 57.2) |
| Multiple signs of critical illness[#] | 6/22 | 27.3 (10.7, 50.2) |

*IMCI: Integrated management of childhood illnesses.

[†] CFR: Case fatality ratio.

[‡] Fast breathing is defined as a respiratory rate of $\geq$ 60 breaths per minute.

[§] Young infant presenting with a single sign.

[¶] Clinical severe infection is defined as the presence of any one of the following signs: severe chest indrawing, high body temperature ($\geq$ 38˚C), low body temperature ($<$ 35.5˚C), stopped feeding well, or movement only when stimulated.

[#] Critical illness is defined as the presence of any one of the following signs: convulsions, unable to feed at all, or no movement at all.

outpatient basis because apart from antibiotics, they also require supportive care such as oxygen, fluid therapy, nutrition, management of hypothermia and hypoglycaemia. Second, a young infant with pneumonia had a very low mortality risk and, therefore, can be effectively and safely treated on an outpatient basis as recommended by WHO [18, 23]. Third, young infants having low-mortality risk signs can be treated on an outpatient basis without referral to a hospital, as demonstrated in the previous trials, in which most of them were cured [19, 21, 22]. Fourth, it has been demonstrated in the AFRINEST and SATT studies that most young infants with signs of clinical severe infection were cured with outpatient simplified antibiotics regimens [19, 21, 22]. However, there were around 10% treatment failures, including about 3% deaths or those who deteriorated clinical. We hypothesize that these young infants with a clinical severe infection who did not benefit from simplified antibiotic regimens on an outpatient basis (having moderate-mortality risk signs) might benefit from a shorter duration of inpatient treatment, compared to critically ill young infants who would need a longer inpatient treatment [8]. Thus, our findings provide research hypotheses to improve the current treatment strategies for sick young infants in low-resource settings.

## Strengths and limitations

The strengths of this study included the analysis of data from a large, high-quality, multi-centre trial. Large sample size from five clinical sites representing east, central, and west Africa

**Table 3. Case fatality ratio (CFR) comparison by the place of treatment and by IMCI* classification [23] and recategorisation of signs of clinical severe infection or severe pneumonia on mortality risk (n = 7046).**

| | Young infants who received outpatient treatment | | Young infants who received inpatient treatment | | p-value |
|---|---|---|---|---|---|
| | Deaths/young infants | CFR[†] % (95% CI) | Deaths/young infants | CFR[†] % (95% CI) | |
| **A. IMCI* classification** | | | | | |
| Pneumonia[‡] | 3/1501 | 0.2 (0.1, 0.6) | 1/342 | 0.3 (0.1, 1.6) | 0.738 |
| Clinical severe infection[§] or severe pneumonia[¶] | 88/4685 | 1.9 (1.5, 2.3) | 23/352 | 6.5 (4.2, 9.6) | <0.0001 |
| Critical illness[#] | 18/83 | 21.7 (13.4, 32.1) | 10/83 | 12.1 (5.9, 21.0) | 0.097 |
| **B. Recategorisation of signs of clinical severe infection[§] or severe pneumonia[¶] based on mortality risk** | | | | | |
| **Low-mortality risk signs**[**] | **37/3726** | **1.0 (0.7, 1.4)** | **12/303** | **4.0 (2.1, 6.8)** | **<0.0001** |
| High body temperature (≥ 38˚C)[††] | 11/1383 | 0.8 (0.4, 1.4) | 0/26 | - | |
| Severe chest indrawing[††] | 10/1310 | 0.8 (0.4, 1.4) | 2/19 | 10.5 (1.3, 33.1) | |
| Only fast breathing in 0–6 days old[††] | 16/1033 | 1.5 (0.9, 2.5) | 10/258 | 3.9 (1.9, 7.0) | |
| **Moderate-mortality risk signs**[‡‡] | **51/959** | **5.3 (4.0, 6.9)** | **11/49** | **22.4 (11.8, 36.6)** | **<0.0001** |
| Movement only when stimulated[††] | 1/30 | 3.3 (0.1, 17.2) | 0/1 | - | |
| Stopped feeding well[††] | 9/243 | 3.7 (1.7, 6.9) | 1/8 | 12.5 (0.3, 52.7) | |
| Low body temperature[††] | 15/161 | 9.3 (5.3, 14.9) | 5/21 | 23.8 (8.2, 47.2) | |
| Multiple signs of clinical severe infection[§] | 26/525 | 4.9 (3.3, 7.2) | 5/19 | 26.3 (9.1, 51.2) | |

*IMCI: Integrated management of childhood illnesses.

†CFR: Case fatality ratio.

‡ Pneumonia is defined as fast breathing (respiratory rate of ≥ 60 breaths per minute) in 7–59 days old infants.

§ Clinical severe infection is defined as the presence of any one of the following signs: severe chest indrawing, high body temperature (≥ 38˚C), stopped feeding well, movement only when stimulated, or low body temperature (< 35.5˚C).

¶ Severe pneumonia is defined as fast breathing (respiratory rate of ≥ 60 breaths per minute) in 0–6 days old infants.

# Critical illness is defined as the presence of any one of the following signs: convulsions, unable to feed at all, or no movement at all. For the current analysis, only these three common signs of critical illness were considered.

** Low-mortality risk signs are defined as infants with a case fatality ratio for any sign ≤ 2.0%.

†† Young infant presenting with a single sign.

‡‡ Moderate-mortality risk signs are defined as infants with a case fatality ratio for any sign >2.0%.

increased the external generalizability of the study. This prospective study utilized trained and well-supervised health workers to collect data during follow-up, reducing the risk of selection and misclassification bias often encountered in retrospective chart-review studies [27]. Finally, the outcome was assessed by independent outcome assessors who were not linked with the treatment. There were a few limitations also. First, the diagnosis was based on only clinical signs, and microbiology or radiology was not used. Second, we did not collect information on comorbidities in this study. Finally, a potential limitation could be that the health centre nurses referred critically ill young infants more strongly than the other sick young infants. Although it is a possibility, we believe that because the health workers were very well trained in the study methods and were supervised, it did not happen on a scale to cause a referral bias [27, 36]. Also, ones who were treated on an outpatient basis were randomised to various treatments [37].

## Conclusion

The mortality risk differs with different clinical signs. The young infants with a possible serious infection can be grouped into those with low-mortality risk signs (high body temperature, or

**Table 4. Case fatality ratio (CFR) with the place of treatment by age and by IMCI[*] classification and recategorisation of signs of clinical severe infection or severe pneumonia on mortality risk (n = 7046).**

| | Age 0–6 days (n = 2707) | | Age 7–59 days (n = 4339) | |
| --- | --- | --- | --- | --- |
| | Outpatient treatment | Inpatient treatment | Outpatient treatment | Inpatient treatment |
| | Deaths/ infants (CFR[†]) | Deaths/ infants (CFR[†]) | Deaths/ infants (CFR[†]) | Deaths/ infants (CFR[†]) |
| **A. IMCI[*] classification** | | | | |
| Pneumonia[‡] | Not applicable | Not applicable | 3/1501 (0.2%) | 1/342 (0.3%) |
| Clinical severe infection§ or severe pneumonia[¶] | 62/2300 (2.7%) | 20/316 (6.3%) | 26/2385 (1.1%) | 3/36 (8.3%) |
| Critical illness[#] | 12/44 (27.3%) | 6/47 (12.8%) | 6/39 (15.4%) | 4/36 (11.1%) |
| **B. Recategorisation of signs of clinical severe infection§ or severe pneumonia[¶] based on mortality risk** | | | | |
| **Low-mortality risk signs[**]** | **27/1902 (1.4%)** | **12/279 (4.3%)** | **10/1824 (0.5%)** | **0/24 (0.0%)** |
| High body temperature ($\geq 38°C$)[††] | 8/588 (1.4%) | 0/12 (0.0%)) | 3/795 (0.4%) | 0/14 (0.0%) |
| Severe chest indrawing[††] | 3/281 (1.1%) | 2/9 (22.2%) | 7/1029 (0.7%) | 0/10 (0.0%) |
| Only fast breathing in 0–6 days old[††] | 16/1033 (1.5%) | 10/258 (3.9%) | NA | NA |
| **Moderate-mortality risk signs[‡‡]** | **35/398 (8.8%)** | **8/37 (21.6%)** | **16/561 (2.8%)** | **3/12 (25.0%)** |
| Movement only when stimulated[††] | 1/11 (9.1%) | 0/1 (0.0%) | 0/19 (0.0%) | 0/0 |
| Stopped feeding well[††] | 5/79 (6.3%) | 1/4 (25.0%) | 4/164 (2.4%) | 0/4 (0.0%) |
| Low body temperature[††] | 13/134 (9.7%) | 4/19 (21.0%) | 2/27 (7.4%) | 1/2 (50.0%) |
| Multiple signs of clinical severe infection§ | 16/174 (9.2%) | 3/13 (23.1%) | 10/351 (2.8%) | 2/6 (33.3%) |

[*]IMCI: Integrated management of childhood illnesses.

[†]CFR: Case fatality ratio.

[‡]Pneumonia is defined as fast breathing (respiratory rate of $\geq 60$ breaths per minute) in 7–59 days old infants.

§Clinical severe infection is defined as the presence of any one of the following signs: severe chest indrawing, high body temperature ($\geq 38°C$), stopped feeding well, movement only when stimulated, or low body temperature ($< 35.5°C$).

[¶] Severe pneumonia is defined as fast breathing (respiratory rate of $\geq 60$ breaths per minute) in 0–6 days old infants.

[#]Critical illness is defined as the presence of any one of the following signs: convulsions, unable to feed at all, or no movement at all. For the current analysis, only these three common signs of critical illness were considered.

[**] Low-mortality risk signs are defined as infants with a case fatality ratio for any sign $\leq 2.0\%$.

[††] Young infant presenting with a single sign.

[‡‡] Moderate-mortality risk signs are defined as infants with a case fatality ratio for any sign $>2.0\%$.

severe chest indrawing or severe pneumonia); moderate-mortality risk signs (stopped feeding well, movement only when stimulated, low body temperature or multiple signs of clinical severe infection), or high-mortality risk signs (critical illness). New treatment strategies that consider differential mortality risks for the place of treatment and duration of inpatient hospital treatment could be developed and evaluated based on these findings.

## Supporting information

**S1 Table. Case fatality ratio (CFR) with the place of treatment by study site and by IMCI[*] classification and recategorisation of signs of clinical severe infection or severe pneumonia on mortality risk (n = 7046).**
(DOCX)

**S1 File.**
(PDF)

## Author Contributions

**Conceptualization:** Yasir Bin Nisar, Antoinette Tshefu, Adrien Lokangaka Longombe, Fabian Esamai, Irene Marete, Adejumoke Idowu Ayede, Ebunoluwa A. Adejuyigbe, Robinson D. Wammanda, Shamim Ahmad Qazi, Rajiv Bahl.

**Formal analysis:** Yasir Bin Nisar.

**Investigation:** Yasir Bin Nisar, Antoinette Tshefu, Adrien Lokangaka Longombe, Fabian Esamai, Irene Marete, Adejumoke Idowu Ayede, Ebunoluwa A. Adejuyigbe, Robinson D. Wammanda, Shamim Ahmad Qazi, Rajiv Bahl.

**Methodology:** Yasir Bin Nisar, Antoinette Tshefu, Adrien Lokangaka Longombe, Fabian Esamai, Irene Marete, Adejumoke Idowu Ayede, Ebunoluwa A. Adejuyigbe, Robinson D. Wammanda, Shamim Ahmad Qazi, Rajiv Bahl.

**Writing – original draft:** Yasir Bin Nisar.

**Writing – review & editing:** Yasir Bin Nisar, Antoinette Tshefu, Adrien Lokangaka Longombe, Fabian Esamai, Irene Marete, Adejumoke Idowu Ayede, Ebunoluwa A. Adejuyigbe, Robinson D. Wammanda, Shamim Ahmad Qazi, Rajiv Bahl.

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
