## [Decision Letter · Decision Letter 0]

17 Feb 2021

PONE-D-20-17173

Clinical signs of possible serious infection and associated mortality among young infants presenting at first-level health facilities

PLOS ONE

Dear Dr. Nisar,

Thank you for submitting your manuscript to PLOS ONE. After careful consideration, we feel that it has merit but does not fully meet PLOS ONE’s publication criteria as it currently stands. Therefore, we invite you to submit a revised version of the manuscript that addresses the points raised during the review process.

Specifically, both reviewers have asked for additional methodological details and further information and clarification of some of the results and observations from the study. In addition, they have provide suggestions to better place your submission in context of previously published work. 

We look forward to receiving your revised manuscript.

Kind regards,

Nicola Stead

Senior Editor

PLOS ONE

"YBN and RB are WHO employees. All other authors declare no competing interests."

Reviewers' comments:

Reviewer's Responses to Questions

**Comments to the Author**

1. Is the manuscript technically sound, and do the data support the conclusions?

Reviewer #1: Partly

Reviewer #2: Yes

2. Has the statistical analysis been performed appropriately and rigorously? 

Reviewer #1: Yes

Reviewer #2: Yes

3. Have the authors made all data underlying the findings in their manuscript fully available?

Reviewer #1: No

Reviewer #2: Yes

4. Is the manuscript presented in an intelligible fashion and written in standard English?

Reviewer #1: Yes

Reviewer #2: Yes

5. Review Comments to the Author

Reviewer #1: Thank you for the invitation to review the study „Clinical signs of possible serious infection and associated mortality among young infants presenting at first-level health facilities“. The study estimates the case fatality rate of different signs of possible severe bacterial infections in infants. The authors found that the mortality risk differs with clinical signs and young infants with serious infection can be grouped into those with lower mortality risk signs; higher mortality risk signs; or signs of critical illness and that only for more severely ill children, inpatient treatment is advantageous compared to outpatient treatment. This is a clinically relevant work that can inform future guidelines on management of PSBI and a well-written manuscript.

I have the following comments:

1. Introduction: “Unfortunately, in many low-resource settings families may not accept referral advice, and sick young infants are not taken to hospital (9-12).” Is refusal by families the major reason for which young infants are not admitted or are there any further barriers to be considered?

2. Methods: „For this analysis, 7129 young infants classified as pneumonia, severe pneumonia, clinical severe infection, or critical illness by IMCI trained nurses at first-level health facilities were selected“ Patients were selected in the outpatient setting and IMCI nurses were trained at first-level health facilities, correct? This sentence is somewhat confusing and the setting of patient inclusion should be better specified.

3. Methods – Statistical analyses: Please be more specific on the analyses conducted, e.g. which tests were used for between-group comparisons? How was missing data handled? Were infants that were initially treated in the outpatient and later admitted to an inpatient facility excluded or counted as in- or outpatient treated cases?

4. Results: “A total of 18420 young infants 0-59 days old were assessed by IMCI-trained nurses at first-level health facilities at five study sites.” Again this may be misleading in terms of setting of patient screening (community vs first-level health facilities).

5. Results: How was the age and sex distribution of young infants with signs of PSBI? Did the authors collect any data on underlying comorbidity? E.g. add a demographics table.

6. Results: Did the distribution of signs of PSBI differ between <7, >= days old?

7. Results: „Of the 7129 young infants, fast breathing (in 7-59 days old)“. Another 18% had fast breathing (in <7d old) according to Fig. 2. I think it would be clearer to refer the proportions to the infants at risk, which means the <7, >=7 day olds or to report a combined proportion of fast breathing.

8. Results: „Compared to inpatient treatment, young infants who received outpatient treatment had a CFR 1.5 times lower for pneumonia (p = 0.74), 3.4 times lower for clinical severe infection or severe pneumonia (p < 0.0001) and 1.8 times higher for critical illness (p = 0.097).“ Were there any differences by study site, level of health facility, underlying comorbidity (if such data is available) or age group of infants (<7 days, >=7 days)? I think it would be important looking a bit deeper into this as this finding may have important implications for care.

9. Discussion: „This finding could be due to several reasons, such as delay in reaching the hospital after accepting referral advice, relatively sicker patients accepting referral advice, delay in getting appropriate treatment after reaching the hospital or inadequate quality of care, and development of nosocomial infections (23-25).“

9.a. Are there any other studies from LMIC with similar results?

9.b. Is there a way to adjust for severity of disease and comorbidities in the data? I think if the authors consider differences in disease severity as a possible explanation of their findings, it would be important to control for this confounder. Otherwise the results should be presented with more caution and this issue should be more prominently included in the discussion.

9.c. Where there any other signs detected by the trained nurses that can help to distinguish infants likely to benefit from hospital admissions vs. infants that may have better outcomes if treated in the outpatient setting?

10. Discussion: Fourth, infants categorized as having higher mortality risk signs, who are not as sick as critically ill, could benefit from a shorter stay in the hospital when they accept referral instead of the recommended seven days (8).

Based on which data was this conclusion made? Please underpin this hypothesis with data or references.

11. Can any conclusions on the definition of sepsis in LMIC be drawn from this data?

Reviewer #2: Abstract page 2:

1) The authors should avoid using abbreviations in the abstract (see Submission Guidelines of the Journal).

Introduction:

1) The authors state on page 4 row 1-6: “Neonatal mortality has decreased…for case fatality rate (CFR) of 9.8% (4).” Please try to supply the reader with more recent data about neonatal mortality, incidence of bacterial infections / sepsis and who they are defined!!! Some of the references are from 2009 and 2012.

2) There is discrepancy between the objective mentioned in the end of the introduction part of the manuscript (page 5 row 7-10) and the objective mentioned in the abstract in the background (page 2, row 3-4) part. The authors should give us consistent description of the “objective of the study”!!!

See the following information which should be integrated in your paper.

Neonatal Mortality ( Reference: data.unicef.org)

The first 28 days of life – the neonatal period – is the most vulnerable time for a child’s survival. Children face the highest risk of dying in their first month of life at an average global rate of 17 deaths per 1,000 live births in 2019, down by 52 per cent from 38 deaths per 1,000 in 1990.

In comparison, the probability of dying after the first month and before reaching age 1 was estimated at 11 deaths per 1,000 and the probability of dying after reaching age 1 and before reaching age 5 was estimated at 10 deaths per 1,000 in 2019.

Globally, 2.4 million children died in the first month of life in 2019 – approximately 6,700 neonatal deaths every day – with about a third of all neonatal deaths occurring within the first day after birth, and close to three-quarters occurring within the first week of life.

Definitions of indicators: Neonatal mortality rate:Probability of dying during the first 28 days of life, expressed per 1,000 live births. Infant mortality rate: Probability of dying between birth and exactly 1 year of age, expressed per 1,000 live births. Under-five mortality rate:Probability of dying between birth and exactly 5 years of age, expressed per 1,000 live births.

Methods:

1) The authors should explain why they defined outcome (“survival status”) on “day 15 after the initial assessment” and not in the end of the neonatal period day 28?

2) Page 6 row 18-21: “CFR, was defined at the…..alone or in combination” This description of CFR definition should be better placed in the Methods part of the manuscript but not in the section of “statistical analysis”.

Results:

1) The authors should decide how they want to present their results. It is unnecessary to present the same information in the text with a written description and a Figure which presents the same information. Therefore, I would suggest to omit Figure 1 from the manuscript.

2) The authors mentioned that 7086 children were included to the study. But on page 7, row 17; “Of the 7192 young infants….” Please clarify this discrepancy in the numbers of patients.

3) The authors should decide once again, how they want to present their results. It is unnecessary to present the same information in the text with a written description (page 7, row 21-23 and page 8, row 1-4) and a table (Table 1) which presents the same information. In this case I would suggest to leave Table 1 which supplies well-arranged information and only “refer the reader” to this table in the results part without presenting the same result in written way.

Discussion:

1) The authors state on page 9, row 1-2 that there was a higher CFR in outpatients compaired to inpatients with the same signs of critical illness this should be more in detail elaborated.

6. PLOS authors have the option to publish the peer review history of their article (what does this mean?). If published, this will include your full peer review and any attached files.

Reviewer #1: No

Reviewer #2: No

---

## [Author Response · Author response to Decision Letter 0]

26 Feb 2021

Point by point to reviewers' comments is attcahed

---

## [Decision Letter · Decision Letter 1]

4 May 2021

PONE-D-20-17173R1

Clinical signs of possible serious infection and associated mortality among young infants presenting at first-level health facilities

PLOS ONE

Dear Dr. Nisar,

Thank you for submitting your manuscript to PLOS ONE. After careful consideration, we feel that it has merit but does not fully meet PLOS ONE’s publication criteria as it currently stands. Therefore, we invite you to submit a revised version of the manuscript that addresses the points raised during the review process.

We look forward to receiving your revised manuscript.

Kind regards,

Emma Sacks

Academic Editor

PLOS ONE

Journal Requirements:

Additional Editor Comments (if provided):

Thank you for your careful revisions and an overall impactful analysis with important practice implications.

In addition to the final additional comments from one reviewer, can you also please address the following?

-Can you provide a bit more information about the distances, especially from first level facilities to hospitals? This might give more context about why families refuse referral and why the journey itself might be more dangerous for critically ill neonates. It might be important for understanding the differences between the 5 sites as well, which would be nice to include in the paper.

-I believe what you describe is actually a case fatality RATIO, not a rate (as it has no time component) ? But I leave it to your judgement

-Please be more specific about the exclusion criteria as related to "missing values" - was this any value, a certain indicator, a set of variables?

-In the ethical approval section, please list the IRBs at each site/country

-Unlike AFRNIEST, which is explained, SATT is not described (nor is the acronym spelled out). If references to SATT are included in the discussion, please give a bit more detail about the SATT trial.

-In the discussion, please include something about the potential of referral bias (or, why you believe there was none?). Could it be possible that nurses more strongly encourage referral of young infants who are more critically ill? These potential biases should also be alluded to in the limitations.

-Please add captions to figures (it looks like there are 2 figure 1s?)

-The paper would benefit from a careful read - there are a few sentences where the technical writing could be improved and abbreviations (such as "neonates up to 28 days were 6040" which is unclear or the contraction "we didn't" could be written out)

-In the future, please use line numbers to assist reviewers in providing feedback.

Reviewers' comments:

Reviewer's Responses to Questions

**Comments to the Author**

1. If the authors have adequately addressed your comments raised in a previous round of review and you feel that this manuscript is now acceptable for publication, you may indicate that here to bypass the “Comments to the Author” section, enter your conflict of interest statement in the “Confidential to Editor” section, and submit your "Accept" recommendation.

Reviewer #1: All comments have been addressed

2. Is the manuscript technically sound, and do the data support the conclusions?

Reviewer #1: Yes

3. Has the statistical analysis been performed appropriately and rigorously? 

Reviewer #1: Yes

4. Have the authors made all data underlying the findings in their manuscript fully available?

Reviewer #1: Yes

5. Is the manuscript presented in an intelligible fashion and written in standard English?

Reviewer #1: Yes

6. Review Comments to the Author

Reviewer #1: I thank the authors for considering my comments and this good revision. The additional analyses increase the value of the study and help to better understand the data. I have only minor comments:

Introduction: The focus of the study is on infants and neonates, thus I would suggesting adding also some epidemiological data on deaths due to infectious diseases/PSBI infection in infants to the first paragraph, which is now presenting data exclusively on neonates.

Methods: „Place of treatment was categorized into either hospitalized or outpatient.“ Place of treatment should be hospital not hospitalized from my point of view.

Table 4: Low body temperature – age 7-59 day, hospitalized treatment: 1/1 should be 100%?

7. PLOS authors have the option to publish the peer review history of their article (what does this mean?). If published, this will include your full peer review and any attached files.

Reviewer #1: No

---

## [Author Response · Author response to Decision Letter 1]

23 May 2021

Rebuttal letter

Point by point response to academic editor and reviewers' comments

PONE-D-20-17173R1: Clinical signs of possible serious infection and associated mortality among young infants presenting at first-level health facilities

Academic editor

Comment#1

Authors' response

We have updated the reference list as per journal requirements in the revised manuscript. 

Comment#2

Can you provide a bit more information about the distances, especially from first-level facilities to hospitals? This might give more context about why families refuse the referral and why the journey itself might be more dangerous for critically ill neonates. It might be important for understanding the differences between the 5 sites as well, which would be nice to include in the paper.

Authors' response

Thank you for this comment. We have added the following information in the revised manuscript: 

"In the Democratic Republic of Congo (DRC), the distance between the health centre (primary health care facility) and the Health Zone General Referral Hospital is approximately 50-55 kilometres (km) in the study area. Since there is no public transportation system available in the study area, people typically walk or ride a bicycle to reach the hospital. The time to reach the hospital can take up to 12 hours by foot or seven hours by bicycle, depending upon the weather, as in the rainy season, it takes a much longer time. In Kenya, the distance between the health centres and the district hospital is around 30 km in the study area. It usually takes several hours on foot and about 50 minutes by a motorized vehicle. In Ibadan, Nigeria, the secondary level referral hospital is approximately 40 km from the Primary Health Centres (PHC) in the study area. It usually takes 90 minutes to reach the hospital through public transport. In Ile-Ife, Nigeria, the referral hospital is situated 50 km away from the PHCs in the study area. The duration of travel time varies from 30 minutes to an hour, depending on the mode of transportation. In Zaria, Nigeria, the Gambo Sawaba General Hospital is at a distance of 15 km from the PHCs in the study area and the Ahmadu Bello University teaching hospital is over 30 km away. Most families walk, while only a few have the privilege of using motorized vehicles as most feeder roads are not motorable, especially during the rainy season. 

Comment#3

I believe what you describe is actually a case fatality RATIO, not a rate (as it has no time component)? But I leave it to your judgement

Authors' response

Thank you for pointing out this. We have changed it to case fatality ratio in the revised manuscript. 

Comment#4

Please be more specific about the exclusion criteria as related to "missing values" - was this any value, a certain indicator, a set of variables?

Authors' response

We have revised this statement in the revised manuscript as given below:

"Cases with missing information about the survival status were excluded from the analysis." 

Comment#5

In the ethical approval section, please list the IRBs at each site/country

Authors' response

We have added the following sentence in the revised manuscript:

"We obtained the site-specific ethics approvals from the University of Kinshasa School of Public Health Ethics Committee, DRC; the Moi University and Moi University Teaching Hospital Institutional Research and Ethics Committee, Eldoret, Kenya; the University of Ibadan/University College Ibadan Hospital Ethics Committee, Ibadan, Nigeria; the Obafemi Awolowo University Teaching Hospitals Complex Ethics Committee, Ile-Ife, Nigeria and the Ahmadu Bello University Teaching Hospital Ethics Committee, Zaria, Nigeria before the enrolment took place."

Comment#6

Unlike AFRNIEST, which is explained, SATT is not described (nor is the acronym spelled out). If references to SATT are included in the discussion, please give a bit more detail about the SATT trial.

Authors' response

Thank you, and we agree with the academic editor. We have mentioned the following two sentences in the discussion section of the revised manuscript and spelled out SATT acronym. 

"Like the AFRINEST, two simplified antibiotic therapy trials (SATT) were conducted in Bangladesh and Pakistan. These trials showed that the simplified antibiotic regimen for young infants with clinical severe infection when referral to a hospital is not feasible was as efficacious as the standard regimen." 

Comment#7

In the discussion, please include something about the potential of referral bias (or, why you believe there was none?). Could it be possible that nurses more strongly encourage referral of young infants who are more critically ill? These potential biases should also be alluded to in the limitations.

Authors' response

Thank you for this suggestion. We have added the following text in the limitation section to address it.

Finally, a potential limitation could be that the health centre nurses referred critically ill young infants more strongly than the other sick young infants. Although it is a possibility, but we believe that because the health workers were very well trained in the study methods and were supervised, it did not happen on a scale to cause a referral bias [27, 36]. Also, ones who were treated on an outpatient basis were randomised to various treatments [37].

Comment#8

Please add captions to figures (it looks like there are 2 figure 1s?)

Authors' response

We have added the appropriate captions to the figures it in the revised manuscript.

Comment#9

The paper would benefit from a careful read - there are a few sentences where the technical writing could be improved and abbreviations (such as "neonates up to 28 days were 6040" which is unclear or the contraction "we didn't" could be written out)

Authors' response

Thank you for this comment. We have read the manuscript carefully and made the necessary corrections to the revised manuscript. 

Comment#10

In the future, please use line numbers to assist reviewers in providing feedback.

Authors' response

We have added line numbers in the revised manuscript. 

Reviewer's comments

Reviewer #1: I thank the authors for considering my comments and this good revision. The additional analyses increase the value of the study and help to better understand the data. I have only minor comments:

Comment#1

Introduction: The focus of the study is on infants and neonates; thus, I would suggesting adding also some epidemiological data on deaths due to infectious diseases/PSBI infection in infants to the first paragraph, which is now presenting data exclusively on neonates.

Authors' response

We had given the following sentence in the first paragraph of the manuscript, “An estimated 9.8% died of the 6.9 million neonatal cases of possible serious bacterial infection (PSBI) in 2012”, which we have modified as follows:

“An estimated 9.8% died of the 6.9 million cases of possible serious bacterial infection (PSBI) in neonates and young infants up to two months of age in 2012”.

Comment#2

Methods: "Place of treatment was categorized into either hospitalized or outpatient. "Place of treatment should be hospital not hospitalized from my point of view.

Authors' response

Thank you for pointing out this. We had used the term ‘hospitalized’ because one could be treated in a hospital as an inpatient or an outpatient basis both. We have modified the place of treatment throughout the revised manuscript as either inpatient or outpatient. 

Comment#3

Table 4: Low body temperature – age 7-59 days, hospitalized treatment: 1/1 should be 100%?

Authors' response

Thank you for this comment. It was a typo error as there were 2 cases of low body temperature in young infants 7-59 days old. We have corrected this in the revised manuscript.

---

## [Editor Report · Decision Letter 2]

31 May 2021

Clinical signs of possible serious infection and associated mortality among young infants presenting at first-level health facilities

PONE-D-20-17173R2

Dear Dr. Nisar,

We’re pleased to inform you that your manuscript has been judged scientifically suitable for publication and will be formally accepted for publication once it meets all outstanding technical requirements.

Kind regards,

Emma Sacks

Academic Editor

PLOS ONE

Additional Editor Comments (optional):

Thank you for your thorough revisions. I believe the findings of this study are actually very important for guiding practice and policy, and I am enthusiastic to see it published. Thank you again for your patience during the review process.
---

## [Editor Report · Acceptance letter]

16 Jun 2021

PONE-D-20-17173R2 

Clinical signs of possible serious infection and associated mortality among young infants presenting at first-level health facilities 

Dear Dr. Nisar:

I'm pleased to inform you that your manuscript has been deemed suitable for publication in PLOS ONE. Congratulations! Your manuscript is now with our production department. 

Kind regards, 

on behalf of

Dr. Emma Sacks 

Academic Editor

PLOS ONE